# pCR and 2-Year Disease-Free Survival: A Combination of the Two Endpoints as a New Classification for Locally Advanced Rectal Cancer Patients—An Updated Pooled Analysis of Eleven International Randomized Trials

**DOI:** 10.3390/cancers15123209

**Published:** 2023-06-16

**Authors:** Maria Antonietta Gambacorta, Giuditta Chiloiro, Carlotta Masciocchi, Silvia Mariani, Angela Romano, Alessandra Gonnelli, Jean-Pierre Gerard, Samuel Ngan, Claus Rödel, Krzysztof Bujko, Robert Glynne-Jones, Johan van Soest, Andre Dekker, Andrea Damiani, Vincenzo Valentini

**Affiliations:** 1Radiation Oncology Department, Fondazione Policlinico Universitario A. Gemelli—IRCCS, 00168 Rome, Italy; mariaantonietta.gambacorta@policlinicogemelli.it (M.A.G.); carlotta.masciocchi@policlinicogemelli.it (C.M.); silvia.mariani@guest.policlinicogemelli.it (S.M.); angela.romano1@guest.policlinicogemelli.it (A.R.); andrea.damiani@policlinicogemelli.it (A.D.); vincenzo.valentini@policlinicogemelli.it (V.V.); 2Radiation Oncology Unit, Azienda Ospedaliero Universitaria Pisana, 56124 Pisa, Italy; alessandra.gonnelli@gmail.com; 3Centre Antoine Lacassagne, 06100 Nice, France; jean-pierre.gerard@nice.unicancer.fr; 4Division of Radiation Oncology, Peter MacCallum Cancer Centre, Melbourne, VIC 3000, Australia; sam.ngan@petermac.org; 5Department of Radiotherapy of Oncology, University of Frankfurt, 60590 Frankfurt, Germany; clausmichael.roedel@kgu.de; 6German Cancer Research Center (DKFZ), 69120 Heidelberg, Germany; 7German Cancer Consortium (DKTK), Partner Site, 60528 Frankfurt, Germany; 8Frankfurt Cancer Institute (FCI), 60596 Frankfurt, Germany; 9Department of Radiotherapy I, Maria Skłodowska-Curie National Research Institute of Oncology, 02-781 Warsaw, Poland; bujko@coi.waw.pl; 10Department of Radiotherapy, Mount Vernon Centre for Cancer Treatment, Northwood, London HA6 2RN, UK; rob.glynnejones@nhs.net; 11Department of Radiation Oncology (Maastro), GROW School for Oncology and Reproduction, Maastricht University Medical Centre+, 6229 ET Maastricht, The Netherlands; johan.vansoest@maastro.nl (J.v.S.); andre.dekker@maastro.nl (A.D.)

**Keywords:** rectal cancer, intermediate endpoints, new risk-based classification, personalized treatment, disease-free survival, pathological complete response

## Abstract

**Simple Summary:**

Locally advanced rectal cancer (LARC) is a heterogeneous disease showing a limited response to neoadjuvant therapies that may be associated with a worse prognosis; therefore, the prediction of long-term outcomes by surrogate endpoints could help with patient stratification from the diagnosis. While the pathologic complete response (pCR) is widely adopted as the primary endpoint in trials, there is little use of 2-year disease-free survival (2yDFS) in this context, although there is some evidence that it can be a strong predictor for overall survival (OS). We conducted a pooled analysis on a large cohort of LARC patients coming from 11 trials, aiming to assess the strength of the combination of pCR and 2yDFS as surrogate endpoints for OS. The results could contribute to understanding the prognostic role of the two single and combined endpoints to identify early on the high-risk patients and consequently tailor therapies.

**Abstract:**

LARC is managed by multimodal treatments whose intensity can be highly modulated. In this context, we need surrogate endpoints to help predict long-term outcomes and better personalize treatments. A previous study identified 2yDFS as a stronger predictor of OS than pCR in LARC patients undergoing neoadjuvant RT. The aim of this pooled analysis was to assess the role of pCR and 2yDFS as surrogate endpoints for OS in a larger cohort. The pooled and subgroup analyses were performed on large rectal cancer randomized trial cohorts who received long-course RT. Our analysis focused on the evaluation of OS in relation to the pCR and 2-year disease status. A total of 4600 patients were analyzed. Four groups were identified according to intermediate outcomes: 12% had both pCR and 2yDFS (the better); 67% achieved 2yDFS but not pCR (the good); 1% had pCR but not 2yDFS; and 20% had neither pCR nor 2yDFS (the bad). The pCR and 2yDFS were favorably associated with OS in the univariate analysis, and 2yDFS maintained a statistically significant association in the multivariate analysis independently of the pCR status. The combination of the pCR and 2yDFS results in a strong predictor of OS, whereas failure to achieve 2yDFS carries a poor prognosis regardless of the pCR status. This new stratification of LARC patients could help design predictive models where the combination of 2yDFS and pCR should be employed as the primary outcome.

## 1. Introduction

Rectal cancer presentation may differ vastly from case to case; in 2009, a stratification of rectal tumors into “good”, “bad”, and “ugly” groups was proposed by Smith according to their MRI characteristics at diagnosis: The “bad” and the “ugly” were considered at an increased risk of distant and distant/local relapse, respectively, and was, therefore, deserving of neoadjuvant treatment [1]. The stage at diagnosis is the most important predictor of survival and metastases remain the main cause of mortality [2]. However, patients exhibit variable responses to neoadjuvant therapies; about 20% of locally advanced rectal cancer (LARC) patients reach a complete response, indicating different tumor biologies [3,4]. The preoperative treatment allowed for a remarkable improvement in local and distant metastasis control without a significant advantage in terms of overall survival (OS) in most trials, but metanalyses and population-based studies could bring out such a benefit in OS [5,6].

OS is considered the endpoint of choice for the objective evaluation of treatment efficacy in cancer clinical trials [7]; yet a large sample size and long and costly follow-up are required in order to detect a survival benefit in a population, potentially exposing confounding factors. The identification of early surrogate endpoints for survival, which in turn can be predicted by statistical models, could lead to the intensification of treatment for those patients predicted to be poor survivors.

Surrogate endpoints are gaining interest in cancer trials in the early prediction of clinical outcomes: Fokas et al. recently reported on surrogate and clinical endpoints used in multimodal clinical trials mainly focusing on the neoadjuvant setting for the standardization of outcome measures in the context of LARC [8]. Currently, the pathological complete response (pCR) is used as the primary endpoint in several trials and was found to have a significant survival benefit [9]. The disease-free survival (DFS) instead is a suitable surrogate endpoint, particularly in adjuvant treatment trials; the DFS at 2–3 years was found to accurately predict OS results at 5–6 years in adjuvant colon cancer studies [10]. Both surrogate endpoints were investigated in a previous pooled dataset of rectal cancer patients in the prediction of OS [11]. The 2-year DFS (2yDFS) was tested against pCR as the surrogate intermediate endpoint and was shown to be better than pCR as a single endpoint, independently of the pCR status, in the prediction of OS. However, currently, LARC studies have infrequently adopted 2yDFS as the primary or intermediate endpoint of OS [8].

Large resources and efforts are required to conduct clinical trials that include a homogeneous population; therefore, combining data from several randomized studies can help draw robust conclusions on a given issue. We, therefore, conducted a pooled analysis with the aim to assess the strength of pCR and 2yDFS as surrogate endpoints for OS on a larger dataset of LARC patients coming from 11 randomized trials and to evaluate if a new risk stratification could be proposed based on the combination of these two endpoints.

## 2. Materials and Methods

### 2.1. Search Strategy and Selection Criteria

Pooled and treatment subgroup analyses were performed on 11 large international rectal cancer randomized studies. We selected from the database LARC patients who received neoadjuvant long-course radiotherapy (RT); concomitant and/or adjuvant chemotherapy (CT) was an option. All patients underwent major surgery; surgical procedures were categorized as low anterior resection (LAR) and abdomino-perineal resection (APR). The absence of complete information on pathological tumor response or treatment, the presence of metastases at diagnosis, and a follow-up time of less than two years in the absence of events (death, local, or distant recurrence) were considered exclusion criteria. Examined trials included: German trials CAO/ARO/AIO-94 comparing preoperative with postoperative chemoradiation and CAO-ARO-AIO-04 evaluating the addition of oxaliplatin to preoperative chemoradiotherapy and postoperative chemotherapy [12,13]; Polish trial I, comparing chemoradiation with preoperative short-course radiation [14], and Polish trial II, comparing chemoradiation to short-course radiation plus consolidation chemotherapy [15]; EORTC22921 trial examining the addition of preoperative or postoperative chemotherapy to neoadjuvant radiotherapy [16]; French trial FFCD9203 comparing preoperative radiotherapy with chemoradiotherapy [17]; TROG01.04 trial comparing short-course and long-course neoadjuvant radiotherapy [18]; Accord 12/0405 trial comparing two different preoperative chemoradiotherapy regimens in T3-T4 rectal cancer [19]; I-CNR-RT trial evaluating the addition of adjuvant chemotherapy after neoadjuvant chemoradiation [20]; Chronicle trial evaluating post-operative chemotherapy versus control [21]; and INTERACT trial comparing two intensification regimens for preoperative CRT [22]. A summary of these randomized trials is reported in Appendix A. Surgical interval (SI, weeks) was calculated from the end of neoadjuvant treatment to the date of surgery.

### 2.2. Outcomes

The pCR was defined as the absence of tumor cells in the resected specimen (ypT0N0). Local recurrence (LR) was defined as tumor presence in the pelvis or perineum documented by imaging or biopsy with or without distant failure. Distant metastasis (DM) was defined as evidence of extra-pelvic tumor tissue confirmed by at least two imaging exams, independent of local status. Cumulative incidence of both LR and DM was reported separately (Appendix A). Survival outcomes were defined as the time elapsed from the end of neoadjuvant radiotherapy to the date of the event; in the absence of the event, the date of the last follow-up examination was considered. Local recurrence and distant recurrence were considered as events for the purpose of DFS; the absence of both LR and DM within two years from the end of treatment was defined as 2yDFS. Death from any cause was the event defining OS. Our analysis focused on the evaluation of OS with respect to pCR status and disease status at two years of follow-up, which was previously recognized as a critical follow-up period [11]. Additional variables were collected concerning clinical characteristics of patients and treatment data.

### 2.3. Data Analysis

We performed an evaluation of the heterogeneity of treatment and clinical characteristics of the population and their impact on various outcomes.

Clinical variables (sex, age, clinical) (c) tumoral and nodal stage (TNM 7th edition), treatment variables (radiotherapy dose, concomitant and adjuvant chemotherapy information, surgical technique, surgical interval, post-treatment pathological (yp) tumoral and nodal stage), long (5yOS) and intermediate (pCR, 2yDFS) endpoints included in the study were first analyzed by descriptive statistic techniques.

Univariate and multivariate logistic regression models were used to identify clinical and treatment factors that could predict 5-year OS for the entire population.

The following analyses were then performed:

Treatment sub-group analysis: We divided the patients into different treatment groups based on the type of therapy they received, including only radiotherapy (nRT), radiotherapy with additional concomitant chemotherapy (nCRT), radiotherapy with both concomitant and adjuvant chemotherapy (nCRT-adjCT), and radiotherapy with additional adjuvant chemotherapy (nRT-adjCT).

Further stratification based on clinical staging: The population was divided into subgroups based on clinical staging, specifically cT3 and cT4 stages.

Investigation of outcome trends: We examined OS trends between patients who did not achieve pCR and between patients who did not achieve 2yDFS and those who did across the different treatment and staging subgroups.

Stratification based on enrolment period: To determine if the enrolment period could influence the impact of intermediate outcomes on OS, patients were further divided into groups based on their enrolment dates (1993 to 1998, 1999 to 2003, 2004 to 2008, and 2009 to 2015).

Identification of subpopulations based on combined pCR and 2yDFS status: Four subpopulations were identified based on their combined pCR and 2yDFS status; these included patients with pCR and 2yDFS, patients without pCR and without 2yDFS, patients with pCR and without 2yDFS, and patients without pCR and with 2yDFS [11].

Evaluation of the impact of intermediate endpoints on OS: We assessed the combined impact of the two intermediate endpoints (pCR and 2yDFS) on OS in the subpopulations stratified based on type of treatment, clinical staging, and enrolment period.

The heterogeneity of different patient subgroups in terms of clinical and treatment variables was assessed by Chi-square or by Fisher’s exact test (as appropriate) and Mann–Whitney test for categorical and numerical variables, respectively. The trend of patients stratified by intermediate outcomes was analyzed using Kaplan–Meier survival analysis with the log-rank test.

The statistical analysis was performed using R version 3.3.1 (CRAN^®^, R Core 2022) [23] and a *p*-value less than 0.01 was considered statistically significant.

## 3. Results

Overall, 7936 patients from 11 randomized studies were considered, 4600 (58%) satisfied the inclusion criteria and were considered in the pooled database. A total of 3336 (42%) patients were excluded for one or more reasons: 2211 of them (28%) did not meet the treatment inclusion criteria, 2913 (37%) had incomplete information on pathological tumor response or treatment, 59 (0.74%) were metastatic at diagnosis, 350 (4%) had a follow-up time of less than two years and were alive in the absence of local or distant relapse (Figure 1).

Information on the clinical and treatment characteristics of the pooled population and the treatment subgroups is provided in Table 1. The median age of the pooled set was 62 years (range 22–82) and 3164 were males (68.8%). At the time of diagnosis, 3923 patients (85.3%) had cT3-T4 rectal cancer and 1818 (39.5%) were node-positive. Within the pooled population, 252 (5.5%) patients were treated with nRT and 1295 (28.2%) with nCRT; in 608 (13.2%) cases, nRT-adjCT was administered, while 2445 (53.2%) patients were treated with nCRT-adjCT. The median overall neoadjuvant treatment duration was 37 days (range 28–40), and the median surgical interval was 43 days (range 28–300). RT was delivered with a median dose of 48.5 Gy (range 26.4–61.2 Gy). Sphincter-saving surgery was performed in 3137 (68.2%) patients. Neoadjuvant and adjuvant CT was administered to 81.3% and 66.4% of patients, respectively; a single-agent fluoride-based regimen was administered in 77.9% and 81.8% in the neoadjuvant and adjuvant settings, respectively. The median follow-up time was 51 months (range 2–213).

Table 2 summarizes the pCR rates and the 2yDFS and 5yOS estimates for the pooled population and treatment subgroups. Among the pooled population, 589 (12.8%) obtained a pCR and 3652 (79.4%) were disease-free at 2 years; the 5-year-OS was 69.3%. In the entire cohort, the 2-year cumulative incidence rates of LR and DM were 6% and 20%, respectively.

For the subgroup of patients who did not achieve 2yDFS, a dedicated analysis was performed in order to detail the incidence of LR and DM at 2 years. The total number of patients without 2yDFS was 948 patients: The incidence of both LR and DM, only LR, and only DM at 2 years were 18%, 8%, and 52%, respectively, while there is no detailed information on 208 (22%) patients. Furthermore, 34 (3%) patients with pCR did not have a 2-year DFS and 88% had DM at 2 years.

In the univariate logistic analysis, pCR, RT dose, adjuvant CT, 2yDFS, and sphincter-saving surgery were found to be favorable variables for 5yOS, while concomitant CT, male sex, age, and both tumor and nodal pathological staging were associated with poorer 5yOS (*p* < 0.001) (Table 3). In the multivariate logistic analysis, RT dose, adjuvant CT, 2yDFS, and sphincter-saving surgery maintained a statistical significance as favorable variables, while age and pathological tumor staging were unfavorable for 5yOS (*p* < 0.001) (Table 3).

Finally, the correlation of the two intermediate endpoints to OS was analyzed. In terms of Kaplan–Meier analysis, both pCR and 2yDFS were favorably related to OS in the pooled population (*p* < 0.001). The same analysis was performed by stratifying the population based on the treatment group: this advantage in terms of OS was maintained and statistically significant, except for pCR in the nRT (*p* = 0.313) and nRT-adjCT (*p* = 0.242) groups (Appendix A). Stratifying patients for the clinical tumor stage (cT3 and cT4) confirmed the association except for pCR in the cT4 subgroup (*p* = 0.082) (Appendix A).

Four groups of patients were identified according to the pCR and 2yDFS intermediate endpoints: 12% (*n* = 546) achieved both pCR and 2yDFS; 67% (*n* = 3106) did not have a pCR but were disease-free after 2 years, 1% (*n* = 34) had pCR but also a recurrence; and 20% (*n* = 905) had no pCR or 2yDFS. Figure 2 shows the overall survival plots for the population and treatment subgroups stratified by combining the pCR and 2yDFS status: The achievement of both pCR and 2yDFS correlates with the best OS trend (the better), followed by the association of 2yDFS and non-pCR (the good), while the patients who did not achieve 2yDFS, regardless of the pCR status, have the worst prognosis (the bad). The advantage holds for both staging groups (cT3 and cT4) (Appendix A).

We also investigated the relationship between 5yOS and the two intermediate endpoints (pCR and 2yDFS) in relation to enrolment time, dividing the population into four groups: accrual time from 1993 to 1998 (*n* = 927, 20.2%), 1999 to 2003 (*n* = 1053, 22.9%), 2004 to 2008 (*n* = 1405, 30.5%), and 2009 to 2014 (*n* = 739, 16.1%); for 476 (10.3%) patients, this information was not known. The pCR and 2yDFS were associated with a statistically significant better 5yOS for all subpopulations (*p* < 0.001) (Appendix A).

## 4. Discussion

The findings from this pooled analysis confirm that the combination of pCR and 2yDFS is a strong predictor of OS in a large population, which has already been pointed out in a previous analysis [11]. Considering the two intermediate endpoints (pCR and 2yDFS), we identified four clinical and prognostic subgroups, where the survival curves showed that the disease-free status at two years confers a stronger gain in 5yOS than pCR: The most convenient situation is when both 2yDFS and pCR have been obtained (the better), whereas not having achieved either endpoint confers the worst prognosis. This finding is in agreement with the results of the Lyon R90-01 trial, where, in the patients achieving pCR, a positive trend in survival was described, which is, however, influenced by disease recurrence at two years [24]. The survival curves for the patients with recurrence within two years were similar regardless of pCR status (the bad); this suggests that when there is an early recurrence, having achieved pCR does not confer a survival benefit. In contrast, patients who achieve 2yDFS have a good prognosis, also when pCR is not achieved (the good). Since DFS includes both local and distant recurrences, it is reasonable to assume that it is a more reliable marker of the tumor complexity and the long-term survival outcomes consequently; it could also be interesting in the future to separately analyze the contribution of the local and distant recurrences with respect to survival and their relation with treatment response.

The clinical tumor stage (cT3 and cT4) and treatment schedule (nRT, nCRT, nCRT-adjCT, and nRT-adjCT) appear not to influence this benefit. We found a slightly different pattern only in the subgroup receiving radiotherapy alone; in this subset, even pCR alone was not shown to be favorably related to OS (*p* = 0.3). This may be attributed to the small sample (*n* = 252, the least represented subgroup) and maybe to worse general clinical conditions of patients not eligible for chemotherapy. Furthermore, the different prognostic value of the two intermediate endpoints was maintained over the 20 years of patients’ enrolment; however, only 2yDFS maintained a statistically significant association with OS in the multivariate analysis.

Smith proposed that rectal cancers that looked “bad” and “ugly” in the preoperative staging should be started on neoadjuvant therapy [1]; we suggested the possibility of a new classification of LARC based on tumor behavior after neoadjuvant treatment: The “better” (those who achieve pCR and are disease-free at two years, with a very good long-term prognosis), the “good” (who achieved 2yDFS but not pCR and have a slightly worse prognosis), and the “bad” (those who experience recurrence within two years, with or without pCR, and have a considerably worse prognosis). The 2yDFS has not currently been explored as the primary endpoint in phase III clinical trials in rectal cancer while the 3-year-DFS has now been accepted as a surrogate for OS in resectable colon cancer [8]. On the contrary, pCR is currently used as the primary endpoint in many trials, as it showed an association with improved DFS and OS [9,25]. This evidence has led to new therapeutic approaches, such as increasing the radiotherapy dose and intensifying preoperative chemotherapy [12,16,17,26], or prolonging the surgical interval to increase the pCR rate and watchful waiting management for those who achieved the clinical complete response (cCR) [3,27]. However, patients with pCR still have a residual risk of distant recurrence and the formal validation of pCR as a surrogate for OS is still lacking.

For them to be effective, more intensive treatments and careful vigilance should be directed to patients with a higher risk of developing early distant metastases, thus the identification of this subset of patients is crucial. Both pCR and 2yDFS are post-treatment endpoints; therefore, models could be generated in the future in order to predict early the combination of the two endpoints for providing personalized therapeutic and follow-up strategies. In addition to the clinical and pathological features, other aspects have been explored as prognostic and predictive biomarkers. The quantitative features extracted from the standard-of-care imaging, in the field of radiomics, can help explore tumor heterogeneity and understand their biological behavior: The radiomic features, often combined within nomograms, proved able to predict tumor characteristics, such as grading and staging [28], pCR [29], and long-term clinical outcomes [30]; furthermore, the radiological biomarker variation in response to treatment is capable of predicting the tumor response to nCRT and the propensity to metastasize and identifying early distant relapse [31,32]. This approach, known as “Delta radiomics”, is a constantly evolving field showing a potential benefit for several clinical endpoints in oncology and could be of particular interest if we want to integrate the understanding of tumor response to treatments into the complexity of rectal cancer management.

This study certainly has limitations. The analyzed population, although large, is drawn from fairly old trials whose accrual time spans over 20 years (from 1992 to 2014); the diagnosis (imaging and pathology) and treatment modalities (chemotherapy, radiotherapy, and surgery) could be of a different quality than the current ones, and some potentially useful clinical information (performance status and comorbidities) are lacking given the retrospective nature of this study. We did not investigate the clinical tumor stage as a risk factor for OS because of the large number of nodal statuses not assessed (33.4%), which could depend on the long accrual time and the different imaging availability. For similar reasons, another limitation is the lack of information regarding more recently recognized prognostic factors in LARC, such as certain imaging (tumor substage, distance from circumferential resection margin, and extramural vascular invasion) or histologic features (mucinous aspects, grading, peri-vascular, and peri-neural invasion) that may contribute to segregate different risk groups.

## 5. Conclusions

In conclusion, based on the previous work by Valentini et al. [16], this present study conducted in a larger homogeneous population confirmed that both pCR and 2yDFS are prognostic, with the latter being stronger than pCR for OS. To influence the natural history of the disease, treatments should not be aimed solely at achieving tumor absence in the resected sample but prediction models incorporating 2yDFS and pCR should be designed in order to identify patients at different risk levels and personalize therapies. In this context, it seems important to use 2yDFS as the primary endpoint in LARC trials to promote comparisons between studies and to find the variables influencing this endpoint that could help build robust predictive models to support clinical decisions.

## Figures and Tables

**Figure 1 cancers-15-03209-f001:**
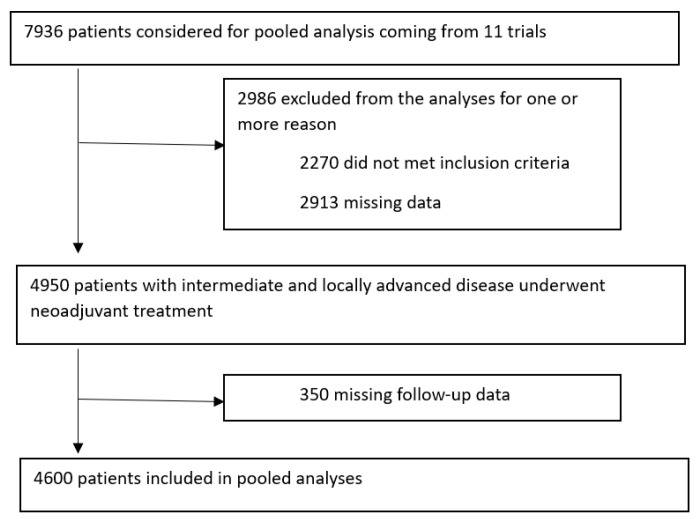
Prisma diagram.

**Figure 2 cancers-15-03209-f002:**
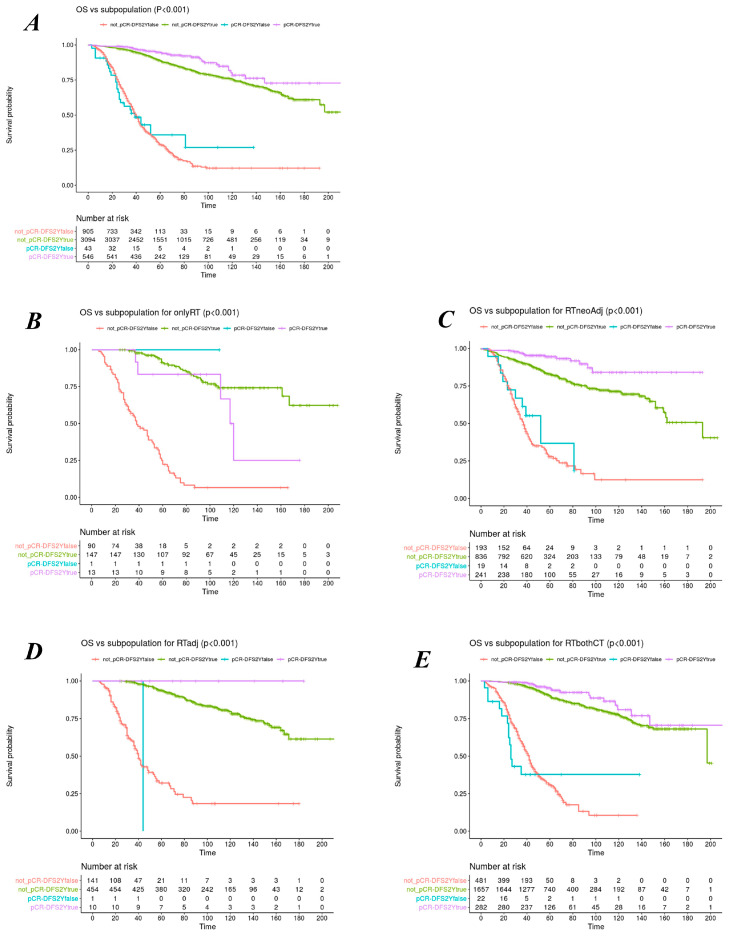
Overall survival plots stratified for combined status of pCR and 2yDFS for the whole population (**A**) and the treatment subgroups of nRT, nCRT, nRT-adjCT, and nCRT-adjCT (**B**–**E**), respectively.

**Table 1 cancers-15-03209-t001:** Clinical and treatment characteristics of the pooled population and the four treatment subgroups.

Dataset	Pooled N	[%]	RT Only N	[%]	RT + Conc.CT N	[%]	RT + Adj.CT N	[%]	RT + Both CT N	[%]	*p*-Value
**Patients**	4600	[100]	252	[100]	1295	[100]	608	[100]	2445	[100]	
**Sex**											
Male	3164	[68.8]	180	[71.4]	878	[67.8]	405	[66.6]	1701	[69.6]	0.374 *
Female	1428	[31]	72	[28.6]	410	[31.7]	203	[33.4]	743	[30.4]	
No data	8	[0.2]	-	-	7	[0.5]	-	-	1	[0]	
**Age** (years)											
Median	62		63		62		60		62		<0.001 **^#^**
Range	22–82		23–79		26–82		30–77		22–82		
**cT stage**											
1 + 2	294	6.47]	0	-	156	[12]	12	[2]	126	[5.1]	<0.001 *
3	3618	[78.6]	214	[84.9]	910	[70.3]	487	[80]	2007	[82.1]	
4	305	[6.6]	27	[10.7]	82	[6.3]	41	[6.7]	155	[6.1]	
No data	383	[8.3]	11	[4.4]	147	[11.3]	68	[11.2]	157	[6.3]	
**cN stage**											
0	1246	[27.1]	16	[6.3]	361	[27.9]	188	[30.9]	681	[27.8]	<0.001 *
+	1818	[39.5]	17	[6.7]	428	[33]	118	[19.4]	1255	[51.3]	
No data	1536	[33.4]	219	[87]	506	[39]	302	[49.7]	509	[20.8]	
**RT dose** (Gy)											
Median	48.5		45		45		45		50.4		<0.001 **^#^**
Range	26.4–61.2		41.4–50.4		28.8–61.2		39.6–55.8		26.4–58		
**Conc.CT**											
5FU/Cape	2680	[58.3]	-	-	949	[73.3]	-	-	1731	[70.8]	<0.001 *
Oxa-based	1059	[23]	-	-	345	[26.6]	-	-	714	[29.2]	
No CT/NA	861	[18.7]	252	[100]	1	[0]	608	[100]	-	-	
**ypT stage**											
0	640	[13.9]	19	[7.5]	269	[20.8]	13	[2.1]	339	[13.9]	<0.001 *
1	303	[6.6]	21	[8.3]	104	[8]	33	[5.4]	145	[5.9]	
2	1331	[28.9]	73	[29]	428	[33]	128	[21]	702	[28.7]	
3	2174	[47.3]	131	[52]	447	[34.5]	413	[67.9]	1183	[48.4]	
4	140	[3]	6	[2.4]	40	[3.1]	18	[3]	76	[3.1]	
No data	12	[0.3]	2	[0.8]	7	[0.5]	3	[0.5]	-	-	
**ypN stage**											
0	3153	[69]	161	[64]	1042	[80.5]	328	[54]	1622	[66.3]	<0.001 *
+	1421	[31]	86	[34]	245	[18.9]	275	[45.2]	815	[33.3]	
No data	26	[0.6]	5	[2]	8	[0.6]	5	[0.8]	8	[0.3]	
**Adj.CT**											
5FU/Cape	2377	[51.7]	-	-	-	-	608	[100]	1769	[72.3]	<0.001 *
Oxa-based	671	[14.6]	-	-	-	-	-	-	671	[27.4]	
No CT/NA	1552	[33.7]	252	[100]	1295	[100]	-	-	5	[0.2]	
**Surgery**											
LAR	3137	[68]	139	[55.2]	874	[67.5]	390	[64.1]	1734	[70.9]	<0.001 *
APR	1463	[32]	113	[44.8]	421	[32.5]	218	[35.9]	711	[29.1]	

* Data are presented as absolute and relative percentage frequencies. The *p*-values were computed by the Chi-square test or the Fisher exact test for categorical and ^#^ Mann–Whitney test for numerical variables. Conc.CT: concomitant chemotherapy; adj.CT: adjuvant chemotherapy; N stage +: node positive; CT: chemotherapy; Gy: Gray; 5FU: 5-Fluorouracil; Cape: capecitabine; Oxa: oxaliplatin; NA: not available; LAR: low anterior resection; APR: abdomino-perineal resection; and nCRT: radiotherapy with additional concomitant CT.

**Table 2 cancers-15-03209-t002:** Outcomes of the pooled population and the four treatment subgroups.

Dataset	Pooled *N* 4600	[100%]	nRT *N* 252	[%]	nCRT *N* 1295	[%]	nRT + Adj.CT *N* 608	[%]	nCRT + Adj.CT *N* 2445	[%]	*p*-Value
pCR											
Yes	589	[12.8]	14	[5.5]	260	[20]	11	[1.8]	304	[12.4]	<0.001 ^#^
No	4011	[87.2]	238	[94.5]	1035	[80]	597	[98.2]	2141	[87.6]	
2yDFS											
Yes	3652	[79.4]	161	[64]	1083	[83.6]	466	[76.6]	1942	[79.4]	<0.001 ^#^
No	948	[20.6]	91	[36]	212	[16.4]	142	[23.4]	503	[20.6]	
5yOS											
Yes	1917	[42]	134	[53.2]	456	[35.2]	410	[67.4]	917	[37.5]	<0.001 ^#^
No	850	[18]	77	[30.5]	256	[19.8]	102	[16.8]	415	[17]	
NA	1833	[40]	41	[16.3]	583	[45]	96	[15.8]	1113	[45.5]	

^#^ Pearson’s Chi-squared test. pCR: pathological complete response; 2yDFS: 2-year disease-free survival; 5yOS: 5-year overall survival; nRT: neoadjuvant radiotherapy; nCRT: radiotherapy with additional concomitant CT; adj.CT: adjuvant chemotherapy; NA: not assessed.

**Table 3 cancers-15-03209-t003:** Univariate and multivariate analysis for 5-year-overall survival according to clinical and treatment-related variables.

Variable	Univariate			Multivariate		
	*p* Value	Odds Ratio	95% CI	*p* Value	Odds Ratio	95% CI
pCR	<0.001	0.386	[0.276–0.530]	-		
RT dose	<0.001	0.943	[0.917–0.969]	<0.001	0.923	[0.883–0.963]
Conc.CT	<0.001	1.485	[1.227–1.803]	-		
Conc.CT Oxa based	<0.001	1.406	[1.111–1.777]	-		
Adj.CT	<0.001	0.690	[0.583–0.817]	<0.001	0.650	[0.503–0.839]
Adj.CT Oxa based	<0.001	1.595	[1.340–1.899]	-		
LAR surgery	<0.001	0.525	[0.444–0.620]	<0.001	0.562	[0.440–0.717]
2yDFS	<0.001	0.035	[0.0279–0.044]	<0.001	0.036	[0.027–0.048]
Age	<0.001	1.013	[1.004–1.022]	<0.001	1.025	[1.013–1.038]
Male sex	0.026	1.223	[1.024–1.465]	-		
cT	0.591	0.957	[0.818–1.123]	-		
ypT	<0.001	1.545	[1.413–1.693]	<0.001	1.30	[1.154–1.477]
ypN	<0.001	2.746	[2.317–3.256]	-		

pCR: pathological complete response; RT: radiotherapy; conc.CT: concomitant chemotherapy; Oxa: oxaliplatin; adj.CT: adjuvant chemotherapy; LAR: low anterior resection; 2yDFS: 2-year disease-free survival. Only variables with a *p*-value < 0.01 in the univariate logistic regression analysis were included in the multivariate model.

## Data Availability

Data used in this study were made available under contact between the different institutes and groups and the University of Sacred Heart, Rome (Italy). The agreements between the European and US institutions were based on the EU General Data Protection Regulation. Requests for datasets should be made to the original investigators from each cohort or trial within the pooled analysis.

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
