# Peer review of "pCR and 2-Year Disease-Free Survival: A Combination of the Two Endpoints as a New Classification for Locally Advanced Rectal Cancer Patients—An Updated Pooled Analysis of Eleven International Randomized Trials"

_cancers, 2023, doi:10.3390/cancers15123209_

Round 1

Reviewer 1 Report

From a biostats and clinical epidemiology point of view, the manuscript has been planned as a pooled analysis of 11 RCT. I have some comments for the Authors:

- all around the manuscript, continuous covariates are to be described as median/IQR and not mean/sd or median/range

- p-values, always with 3-sign digits (p=0.742 or p<0.001), avoiding p<0.01 and so on

- line 120, the estimation of cumulative incidence (CI) is a tricky story, counting on an huge number of references. Mainly, CI requires 2 mutually excluding events. So, this stats approach has to be redone, following what suggested in Bone Marrow Transplant. 2007 Aug;40(4):381-7 and Bone Marrow Transplant. 2010 Sep;45(9):1388-95 (2 very few technical papers). Are you sure that expressing your results as a CI estimation is your goal?

- line 121, the definition of DFS has to be added (several different do exist)

- line 127, stat analysis section has to be totally and properly set up (i.e. descriptive/inferential/modeling/subgroup and so on)

- line 148, Chi-squared test deals with stochastic independence

- line 152, p<0.01 cutoff, why not the more common p<0.05!? Mind all the results at the light of this choice

- line 173 Median overall neoadjuvant treatment time, better to say duration!

- table 1, it's quite messy, why not a simpler one!?

- table 2 dataset, I can't understand the words yes/no for DFS and OS, since they do refer to a time period (time-to-event outcomes)

- line 199 univariate analysis, which one? Cox or logistic?

- line 202 multivariate analysis, see above!

- table 3, report all results as HR, 95%CI and p-value, nothing more nothing less

- line 216 the correlation of the two intermediate endpoints, this is a potential association

- line 216, IMHO there's an excess in subgroup analyses, readers got lost into them

- figure 2, very poor KM curves quality 

- line 307 2yDFS as a reliable predictor for overall survival, stronger than pCR, I do believe that this statement has to be moderated, due to pooled analysis nature of your research, only future RCTs could confirm these preliminary infos

Minor editing

Author Response

We would like to thank the Reviewer for his comments and suggestions. Below you will find the answers point by point.

  • p-values, always with 3-sign digits (p=0.742 or p<0.001), avoiding p<0.01 and so on

We thank the reviewer for this comment; we provided 3-digits p-values.

  • line 120, the estimation of cumulative incidence (CI) is a tricky story, counting on an huge number of references. Mainly, CI requires 2 mutually excluding events. So, this stats approach has to be redone, following what suggested in Bone Marrow Transplant. 2007 Aug;40(4):381-7 and Bone Marrow Transplant. 2010 Sep;45(9):1388-95 (2 very few technical papers). Are you sure that expressing your results as a CI estimation is your goal?

We thank the reviewer for this comment; we agree that it is not a simple concept and it was not the main goal of our analysis, but we would like to provide this additional information and we put curves on cumulative incidence in the supplementary material.

  • line 121, the definition of DFS has to be added (several different do exist)

We thank the reviewer for this comment; we slightly changed the sentence so that it may result clearer.

  • line 127, stat analysis section has to be totally and properly set up (i.e. descriptive/inferential/modeling/subgroup and so on)

We thank the reviewer for the comment; we changed the paragraph accordingly.

  • line 148, Chi-squared test deals with stochastic independence

We thank the reviewer for the comment; we changed the Data analysis paragraph.

  • line 152, p<0.01 cutoff, why not the more common p<0.05!? Mind all the results at the light of this choice

We thank the reviewer for this comment. Given the large number of patients, we decided to adopt a more conservative approach using a p-value<0.01 as the statistically significant cut-off; all results were then reported using this criterion.

  • line 173 Median overall neoadjuvant treatment time, better to say duration!

We thank the reviewer for this comment; we provided changes accordingly.

  • table 1, it's quite messy, why not a simpler one!?

We thank the reviewer for this comment; we provided a simpler table.

  • table 2 dataset, I can't understand the words yes/no for DFS and OS, since they do refer to a time period (time-to-event outcomes)

We thank the reviewer for this comment; we would like to underline that the data refer to DFS and OS estimates at a certain time point by Kaplan Meier method as stated in the Method section, therefore they are presented as numbers-percentages instead of time intervals. We modified the related paragraph accordingly.

  • line 199 univariate analysis, which one? Cox or logistic?

We thank the reviewer for this comment; we used a logistic analysis, as stated in the Methods section.

  • line 202 multivariate analysis, see above!

We thank the reviewer for this comment; we used a logistic analysis, as stated in the Methods section.

  • table 3, report all results as HR, 95%CI and p-value, nothing more nothing less

We thank the reviewer for this comment; we provided changes accordingly.

  • line 216 the correlation of the two intermediate endpoints, this is a potential association

We thank the reviewer for this comment; what we analyzed was the association of the two intermediate endpoints with OS.

  • line 216, IMHO there's an excess in subgroup analyses, readers got lost into them

We thank the reviewer for this comment; we here report the results for the analysis we performed in a fairly concise way, in any case we tried to streamline the paragraph.

  • figure 2, very poor KM curves quality

We thank the reviewer for this comment; we provided sharper figures.

  • line 307 2yDFS as a reliable predictor for overall survival, stronger than pCR, I do believe that this statement has to be moderated, due to pooled analysis nature of your research, only future RCTs could confirm these preliminary infos

We thank the reviewer for this comment; we provided changes accordingly.

Reviewer 2 Report

Dear authors, thank you very much for the opportunity to review your paper: “pCR and 2-year disease-free survival: a combination of the two 2 endpoints as a new classification for locally advanced rectal cancer patients. An updated pooled analysis of eleven international randomized trials”.

The paper investigates pathological complete response after neoadjuvant treatment and 2-year disease free survival as prognostic factors for overall survival and propose a new classification.

The paper is well written and easy to read. The tables present all relevant data. The topic is very interesting. I have some questions and comments:

As there are 6.5% cT1/2 stages, the authors could present a definition of advanced rectal cancer and the indication for neoadjuvant therapy. The prefix “c” should be explained. 

pT0 is not a UICC stage and should be explained. Does it mean no visible tumor? In addition, the suffix “y” should be added.

Why is there a difference between pT0 (13.8%) and pCR 12.8%?

The number at risk for the Kaplan Meier plots could be presented.

The most interesting finding is the poor result for patients with pCR and recurrence after 2 years. This finding could be further discussed. pCR describes the local situation after operation. Despite complete response, a significant number of patients develop metastasis. DFS is a sum of local recurrence and distant metastasis. Maybe it could be a good idea to look at metastasis only. The percentage of local recurrence is not presented. Overall survival is not only dependent on tumorbiology and DFS but also on patients’ performance status. Patients with comorbidity will die of other causes. Do you have data regarding the performance status? 

The description of 2y-DFS is for an indivualised therapy and the patient too late. Therefore, I would not call it a new classification system. You should further explain what you mean with “prognostic models. We know that minimal residual disease or circulating tumor DNA after operation are very strong predictors for recurrence.

Could you further discuss your results in regard of a “watch and wait” strategy? If a significant number of patients with clinical complete response and even pathological complete response develops recurrence, your proposed prognostic modell should be applied and the patient should not be included in a “watch and wait” protocol.

The paper and the results are very interesting, and the paper should be accepted for publication.

Author Response

We would like to thank the Reviewer for his comments and suggestions. Below you will find the answers point by point.

  • As there are 6.5% cT1/2 stages, the authors could present a definition of advanced rectal cancer and the indication for neoadjuvant therapy. The prefix “c” should be explained.

We thank the reviewer for this comment. cT1/2 tumors were node-positive and/or low-located and therefore met the inclusion criteria of the studies; we provided an explanation for the prefix.

  • pT0 is not a UICC stage and should be explained. Does it mean no visible tumor? In addition, the suffix “y” should be added.

We thank the reviewer for this comment. T0 is defined as “no evidence of primary tumor” by the UICC TNM staging classification; we added the prefix as suggested.

  • Why is there a difference between pT0 (13.8%) and pCR 12.8%?

We thank the reviewer for the comment. As stated in line 116, pCR was defined as the absence of tumor cells in the whole resected specimen (ypT0N0), not only in the primary site, that is why the numbers slightly differ from each other.

  • The number at risk for the Kaplan Meier plots could be presented.

We thank the reviewer for the comment; we provided curves with numbers at risk.

  • The most interesting finding is the poor result for patients with pCR and recurrence after 2 years. This finding could be further discussed. pCR describes the local situation after operation. Despite complete response, a significant number of patients develop metastasis. DFS is a sum of local recurrence and distant metastasis. Maybe it could be a good idea to look at metastasis only. The percentage of local recurrence is not presented. Overall survival is not only dependent on tumorbiology and DFS but also on patients’ performance status. Patients with comorbidity will die of other causes. Do you have data regarding the performance status?

We thank the reviewer for the comprehensive comment; we unfortunately lack data on patients’ performance status and comorbidities given the retrospective nature of the pooled analysis. We provided changes in the Discussion section according to his suggestions.

  • The description of 2y-DFS is for an indivualised therapy and the patient too late. Therefore, I would not call it a new classification system. You should further explain what you mean with “prognostic models. We know that minimal residual disease or circulating tumor DNA after operation are very strong predictors for recurrence.

We thank the reviewer for the comment; our future intent is precisely to get to develop a tool to anticipate this risk classification at the time of diagnosis. We modified the related paragraph in the Discussion section in order to make it clearer.

  • Could you further discuss your results in regard of a “watch and wait” strategy? If a significant number of patients with clinical complete response and even pathological complete response develops recurrence, your proposed prognostic modell should be applied and the patient should not be included in a “watch and wait” protocol.

We thank the reviewer for the comment; the discussion of a “watch and wait” strategy is beyond the scope of this study, but we know it is a topic of great relevance currently and we covered this topic in other publications.

Round 2

Reviewer 1 Report

I have no more comments

Minor